# Business Model for Scaling Social Impact towards Sustainability by Social Entrepreneurs

**Kishore Kumar François** [1,*] and **Hoe Chin Goi** [2]

1 NUCB Undergraduate School, Nagoya University of Commerce and Business, Sagamine 4-4, Komenoki-cho, Nisshin 470-0193, Aichi, Japan
2 NUCB Business School, Nagoya University of Commerce and Business, 1 Chome-3-1 Nishiki, Naka Ward, Nagoya 460-0003, Aichi, Japan; goi_hc@gsm.nucba.ac.jp
* Correspondence: kishore_kf@nucba.ac.jp or kishorekf@gmail.com; Tel.: +81-0561-73-3006

**Abstract:** This paper examines a Business Model (BM) from a socio-economic system perspective to discern key factors and understand its interactions resulting in the Scaling of Social Impact (SSI) in Social Entrepreneurship (SE). Previously, studies have explained the importance of the BM in relation to SE. However, there is a lack of empirical studies on how a BM's transitions through participation of various actors result in the SSI, causing a gap in this field's research. This research applies a qualitative analysis on a single case study of a Japanese social startup, "mymizu", the first water refill application platform in Japan. The findings show that collaboration amongst different stakeholders on the initial phase of the BM could increase awareness of responsible consumption, convert into actual users for sustainability, and change their behavior. Secondly, members of society could take on dual roles, both as users and collaborators in the BM, which results in an exponential scaling effect of the Social Impact (SI). This paper contributes towards adding a Participatory Stakeholder (PS) to the ecosystem of the SSI and building a Regenerative BM (RBM) that is relevant in SE towards sustainability.

**Keywords:** business model; scaling social impact; social entrepreneurship; social impact; scale; society

## 1. Introduction

Governments and corporations do not always have the capability to respond to the increasing and diverging needs of the people and the planet; hence, social startups and enterprises are gaining an important role through the innovative solutions they bring to socio-environmental issues [1–4]. As such, there is an increasingly important need for social entrepreneurs to achieve and balance both social and economic outcomes for environmental and business sustainability [5] in a circular economy. With innovative Business Models (BM) and the democratization of technologies and information, there has been a fast growth of the Social Entrepreneurship (SE) phenomenon, which has led to further traction from both academics and practitioners alike, growing the number of literature and case studies around the world [4,6]. Given the vast variety of BM and social missions incorporated in SE, various definitions exist for the concept of SE, further divided into different focus areas and levels, impacted by the trends and times at which they appear [7], and with a variety of stakeholders playing different roles. Adding onto this vagueness is how the BM impacts the flow of SE when it transitions from its founding to growing and scaling (of the organization and of the Social Impact (SI)).

Social entrepreneurs, with their missions grounded in a social purpose, aim to create social value [8] by delivering SI, which is defined as significant or positive changes that solve or address social issues [6]. They achieve this through the BM that diverges from a traditional one, as they are driven by a social mission and not economic pursuit. And in aiming to deliver the highest SI possible, social entrepreneurs face numerous challenges: lack of funding, lack of support and skills, balancing business sustainability and mission, and ultimately, the ability to achieve the Scaling of Social Impact (SSI) [5,9–12]. In overcoming

these constraints, SE can stimulate sustainable development, favoring greater achievement of the Sustainable Development Goals (SDGs) [3,5].

The literature shows that SE should rely on its network of stakeholders and actors and on its ecosystem to overcome these constraints [9–12]. In parallel, a growing body of literature on the BM is also concerned with the systemic perspective, bringing consensus to the importance of incorporating other actors' roles within a BM [13,14]. Consequently, the SSI's research stream posits that it is more pertinent to achieve the SSI in an exponential manner without scaling the organization but instead to focus on the ecosystem and systemic actors [15]. As such, the current literature is developing and converging on the importance of a systemic perspective through various actors and stakeholders. However, these works are developed separately, without establishing inter-relationships between them. In particular, the relationship between a BM and the SSI has not been studied enough, so there is a lack of understanding on how these strands of work might be synthesized to offer insights into a BM's role in the SSI in the context of SE. This research gap is highly significant for both researchers and practitioners due to the increasing scholarly attention to topics related to SE [10] and the rising importance of social entrepreneurs as change agents in solving social and environmental problems [5].

Hence, we establish the following as the research question: what is the role of the BM in the SSI and how does it result in an exponential scaling? To answer this research question, this paper aims to advance efforts by drawing on the literatures of SE, BM, and the SSI to examine the role of the BM and its transitions to the SSI in the context of SE. For this purpose, we adopt a systemic perspective to determine the key factors of a BM in order to understand how they lead to the SSI. Thus, this paper has a unique position to build both theoretical and practical contributions, for academics and social entrepreneurs alike, and leverage the case study research method to investigate the SE phenomenon within its real-world context. A single case study, "mymizu", a Japanese social startup that combats the issue of plastic waste, is applied to analyze these inter-relations in the context of the circular economy in Japan, a highly significant area given the current environmental circumstances and socio-economic revitalization. A theoretical framework is drawn by generalizing the case study's findings. Practical guidelines and implications are also derived on how innovations in the BM would help the SSI for social entrepreneurs. Social entrepreneurs should first focus on their BM's value proposition to communicate clearly to society (particularly the target audience) in order to receive their engagement; they should then establish partnerships with like-minded actors to gain further legitimacy; and finally, they should involve members of society and partners in key roles within the organization. On the other hand, society should also leverage these opportunities to become involved in collective efforts towards a circular economy.

This paper is structured as follows: Section 2 provides a literature review of SE, BM, and the SSI, with a particular emphasis on the systemic perspective to introduce the importance of a network for SE, the activity system approach for the BM [13], and the ecosystem of the SSI [15]. In particular, the latter two papers are considered to be pioneering with their introduction and highlighting of system elements to the research strands of the BM and the SSI, where much of the past efforts regarded organizational factors and growth. Section 3 details the research methodology and introduces the singular case study of "mymizu", the Japanese social startup which has been acclaimed for its social mission. Section 4 presents the results from the qualitative analysis through the case study, and we discuss their implications and contributions in Section 5, updating the current model of the ecosystem of the SSI with a new factor. The conclusion then provides remarks on the limitations of this paper and paves the way for further research to be conducted.

## 2. Literature Review

In this section, we review the literature on core aspects of this paper, SE, BM, and the SSI, highlighting important insights relevant to this study. In particular, we explored recent

studies that have introduced systemic perspectives to call attention to the importance of collaboration and partnerships across various sectors and entities [9–13,15–17].

### 2.1. Social Entrepreneurship

SE became an increasingly important phenomenon in the 21st century, gaining traction amongst both researchers and practitioners for its contribution to environmental and socio-economic justice, development, and prosperity. Social entrepreneurs are considered to be filling in the institutional void unfilled by other entities, viewing social issues as an entrepreneurial opportunity [3,4,10,18]. Nevertheless, given the vast contexts of "social" (which encompasses a much larger area) issues, numerous and ambiguous definitions exist [19]. Amongst existing definitions, a popular one is by Zahra et al. [6], who defined SE as "the activities and processes undertaken to discover, define, and exploit opportunities in order to enhance social wealth by creating new ventures or managing existing organizations in an innovative manner". Amongst recurring themes of SE in the literature, we found "social wealth/value" as the outcome of their mission [20] and "innovation" to define the approach to activities and processes. Indeed, CASE [21] and Dees et al. [22] mentioned that the goal of social enterprises is mostly to maximize their SI through an approach that includes the scaling of the BM, with innovation being a key influence [23] for delivering this SI. With such an important mission in their field of operation, social entrepreneurs are considered as change agents [20]. The European Commission's definition also emphasized the concept of SI, by defining a social enterprise as "an operator in the social economy whose main objective is to have a SI rather than make a profit for their owners or shareholders". Hence, in the context of this paper, we define SE with a focus on the innovative activities and processes undertaken to deliver SI.

In overcoming SE's constraints, more research has emerged recently on the benefits of social entrepreneurs leveraging their ecosystem (composed of networks of actors and a variety of activities) and relying on the interactions of actors and their support activities [10,24]. This communication and engagement create a better support system for social entrepreneurs to tap into external resources, have a better outreach, and to achieve the SSI [11,12]. Kovanen [16] highlighted that community collaboration can enable social entrepreneurs to better balance their institutional and resource relations and to reach societal change, a major SI sought after by entrepreneurs. However, the term community has been loosely used, and the scope remains to be defined more precisely, as collaboration can vary greatly depending on the size of the community. A collaborative SE is perceived to be successful when the process is carried out in a participatory manner [16]. Social startups, with the importance of their social missions, are capable of attracting volunteers through a sense of purpose, enabling them to be part of social change [10,18]. To this extent, it has been shown that collective action frameworks can serve as strategies to drive systems change through innovative methods and motivate supporters to action [25]. This then raises the question as to how the community can be involved.

These developments, when seen through the stakeholder theory's lens, allow us to confirm the relevance of society (encompassing communities and individuals and considered as a network of resources, capabilities, and opportunities by Goduscheit et al. [4]) as a key stakeholder in SE that enhances the delivery of SI. However, there is a gap in the understanding of how such stakeholders can be involved in the BM of SE. Indeed, social entrepreneurs make use of the entire spectrum of legal forms, thus deploying a vast variety of BM to work towards their social mission [26]. This enforces the need for a deeper apprehension of the BM as used by social entrepreneurs. Furthermore, various scholars have demonstrated that managers (social entrepreneurs are also managers within their ventures) interested in social and environmental value creation are using the BM concept more than ever [27]. In particular, social entrepreneurs are poor in resources, thus comes the need to look for innovative BM in order to make their social venture financially sustainable [5].

*2.2. Business Model*

Although a large body of literature exists on the BM with its rising popularity, the definition of a BM remains various (partly impacted by the organizational goals, such as SI, profitability, growth, etc.), with both researchers and practitioners developing their studies according to their purpose and phenomena of interest (such as SE and sustainability, which are popular in recent research developments), thus being unable to use a single language to compare BM frameworks [27,28]. Furthermore, research on the BM has typically focused on the organization itself and its internal systems, and how they created, captured, and delivered value to its customers [29]. However, this last decade saw a growing consensus on how the BM represents "a system of interdependent activities that transcends the focal firm and spans its boundaries" [13]. This brought forward new research strands into holistic and systemic perspectives of the BM concept, going beyond the organization's internal view and boundaries to outline how the BM interacted with its surrounding external environment [30], which became its ecosystem [14] and represented a fundamental characteristic of the BM for sustainability [29]. This direction is crucial in this literature, as social entrepreneurs leverage collaboration, thus requiring a better knowledge of what it means to understand a BM's design and structure.

In Zott and Amit's [13] definition of the BM through an activity system perspective, the set of interdependent activities could be conducted by the organization itself or its partners. As such, although the interdependent activities can go beyond the organization's boundaries, they remain focused on the organization to create and retain a share of the value. However, the activities may also be performed by its partners, enabling the organization to tap into external resources and capabilities, which evokes the idea of operating through an "open BM" [13]. Building on this concept through the lens of open systems theory, Berglund and Sandstrom [30] established that organizations are influenced by their environment, depending on external actors for critical resources despite their unreliability due to being outside the organization's control. This calls for better relationships between the organization and external actors through feedback loops for hedging the uncertainty. Bolton and Hannon [31] took on the same principle, demonstrating that the more successful BM entrepreneurs donned the role of system builders through partnerships in order to draw on resources. Such arguments were echoed by Kovanen's [16] findings on resource relations being important for social entrepreneurs, with collaboration being a key factor to balance them.

Indeed, resulting from the interdependent activities that transcend an organization's boundaries, value creation is carried out through these exchange relationships among multiple players, showcasing that BM as a concept focuses on cooperation, partnerships, collaboration, and joint value creation [28]. Moreover, for entrepreneurs thinking (or rethinking) of their BM design, Zott and Amit [13] argued that a focus on activities was an important perspective and that the activity system perspective encouraged them towards systemic and holistic thinking instead of narrow and isolated choices, which is beneficial in leveraging resources, as previously mentioned.

While the BM's notion of value is often economic, in the context of sustainability, it takes a broader definition to encompass social and environmental aspects. Thus, the triple bottom line approach became a major concept for the BM, highlighting the need to consider stakeholder interests, such as the society and environment [32], by communicating how a BM created and delivered the value [33]. Evans et al. [34] went beyond to say that a sustainable value flow was necessary among multiple stakeholders, including the environment and society as primary stakeholders. Such sustainable value flow can be generated through either cooperation or collaboration between the different business and non-business actors, such as the government or society, possibly paving the way for scaling a (sustainable) BM [17,35]. This direction of research on the BM in the context of sustainability highlighted the systemic perspective that was established by Zott and Amit [13], and it built on it further by going into the concepts of stakeholders, partnerships, and collaborations, which are recurring themes in this paper.

However, scaling a BM (through collaboration as stated by Ciulli et al. [17]) is equivalent to scaling the organization, and while that is one approach for the SSI, this paper focuses more on understanding how to achieve the SSI in an exponential manner by relying on the ecosystem and systemic actors [14]. This goes beyond a more traditional internal and organizational perspective, as both researchers and practitioners are more commonly interested in how such an approach is more beneficial.

### 2.3. Scaling Social Impact

As shown in the literature review of SE, SI is the raison d'être of social enterprises, and social entrepreneurs thrive to create social value for their mission; thus, the SSI becomes a critical phenomenon for social enterprises [22,36,37], to the extent of being considered as "the single most important criterion to judge the performance of social enterprises" [38,39]. This brought much interest, from both researchers and practitioners, into knowing the factors that enable or limit the potential of the SSI [40–42], and research showed that the SSI is indeed one of the most challenging issues in social enterprises [43–45]. Furthermore, with few social ventures experiencing scaling, it becomes one of the most important and least understood topics in SE research [46].

The current literature defined the SSI as an "ongoing process of increasing the magnitude of both quantitative and qualitative positive changes in society by addressing pressing social problems at individual and/or systemic levels through one or more scaling paths" [47]. This was built on the definition of SI given by Zahra et al. [6] with the term "scaling paths", highlighting that there are various approaches to the SSI.

The common approach existing in the literature is grounded on scaling the BM as part of scaling the organization itself [21,22], with the growth of the BM also being conceptualized as scaling [17]. Drawing from the literature of entrepreneurship, the term scaling is used for organizations that go through a "persistently rapid growth" [48], with "scalability" as a related concept that refers to the capacity within the BM to increase sales towards a growing customer base [49], based on replicability, adaptability, and transferability of the operational model as key factors for scalability [44,50]. Dees et al. [22] defined scalability as "increasing the impact a social-purpose organization produces to better match the magnitude of the social need or problem it seeks to address". Therefore, this approach perceives the SSI as organizational growth, with a focus on the organizational factors (internal capacities and capabilities) that make this a reality.

However, it is known that social entrepreneurs face numerous constraints in regard to their resources and capacities, which is why innovation is leveraged as a key factor for their mission to deliver SI [6]. Given this reality, scaling the organization (BM) is a major challenge for social entrepreneurs. Hence, the SSI is more about the effectiveness of addressing the social issue, transforming perspectives on issues, and changing the status quo, rather than just increasing the impact through "persistently rapid growth" [15]. To this extent, it was shown that systemic-level factors are currently understudied in the current literature of the SSI, with Han and Shah [15] suggesting an ecosystem framework that discussed the roles of different stakeholders for business creation and operation that results in the SSI. The holistic approach provided clarity to the different stakeholders' roles and drew a parallel with SE and a BM driven by multiple stakeholders, in contrast to the internal perspective shown by the previous approach. In their research, the authors showed how the current literature did not distinguish the SSI and scaling organizations, and that it was more important and interesting to figure out how to achieve the SSI without maximizing organizational growth. This echoed Bradach [44]'s words of "how to get $100\times$ the results with $2\times$ the organizations", supporting the argument of constraints faced by social entrepreneurs and the need for effectiveness in addressing social issues. In their framework entitled "ecosystem of SSI", they incorporated interconnected key elements such as financing, government policy, institutional infrastructure, and the process of scaling as central elements, with the latter embodying the organizations that use different strategies to achieve the SSI through technology and data. All of these four elements led to SI as

the outcome of their inter-related activities, which made up the whole ecosystem's efforts towards SI. The process of scaling itself can be considered to encompass the BM, with other elements representing part of Demil et al.'s [14] view of the ecosystem and Zott and Amit's [13] view of the BM as an activity system.

From the literature review thus far, we found a gap in the understanding of the BM and its role in the SSI, namely for the approach that does not rely on scaling the BM itself [15]. Demil et al. [14] also argued that both BM and business ecosystems were not static but rather co-evolved, which leads us to explore how their co-evolving transitions influence the outcome: the SSI in the context of SE. While the literature review on the BM and the SSI (and SE) shared the recurring theme of exploring a holistic and systemic perspective in order to include and engage external stakeholders [9–13,15–17], there is a need for empirical evidence of how a BM can incorporate other actors to impact the process of scaling, leading to the SSI. As such, this paper explores the roles of a BM in depth through the analysis of a singular case study, for which we employ Han and Shah's [15] framework.

## 3. Research Methodology

This paper employed a qualitative research method, as it is particularly beneficial in dealing with the nature and complexity of the phenomena (BM and the SSI), investigating them in their natural environment of SE, and reconciling the complexity and details in the context of the study [51–55]. This study was based on an inductive approach that allowed for theoretical development by exploring current theory, data, and the inter-relationships between the variables [56,57]. We employed a single holistic case approach based on a Japanese social startup, mymizu, with successful SSI track records in the context of a circular economy; collected primary and secondary data through direct interviews with key personnel, fieldwork observations, document study, and social media; and applied a content analysis method to formulate the findings and the resulting discussion points.

### 3.1. Case Study Method

We applied an in-depth single case study with longitudinal observation and research from the year 2019 until 2023. The case study approach investigated the contemporary phenomena of the BM and the SSI in depth and within the real-world context of SE to understand the case and how and why it worked [58]. There is also a lack of case studies to showcase how organizations can innovate and design the BM in novel ways in order to work towards sustainability and deliver the highest SI possible [34].

This methodology focused on the process and scaling outcomes of the SI of a Japanese social startup, mymizu, given its position in a niche environment. Mymizu is highly relevant as the single case study for the context of this paper because it is an exemplary social startup with a unique BM which leverages creativity, innovation, digital, and social factors for delivering its social movement, in conjunction with numerous actors. As it operates within the Japanese ecosystem, it echoes aspects found throughout the literature review in regard to being a social startup, its BM, and its SSI.

Amongst papers reviewed in this study's literature review of SE, BM, and the SSI, the qualitative research design was the most commonly employed approach through case studies (often multiple rather than single) for empirical evidence, along with scoping of literature reviews. This can be explained by the fact that much of the literature was focused on providing definitions and building frameworks for SE, BM, and the SSI, with less focus on inter-relationships between these concepts; hence, there was a broader application of this methodology. On the other hand, this study is more in depth, as it focuses on exploring the relationships and interactions between the BM and the SSI within the context of SE. Hence, the dive into the single case study of mymizu enables us to explore the relationships with a deeper understanding [55–57].

### 3.2. Case Study of Mymizu

This award-winning social startup was established in 2019 as a brand under the umbrella of Social Innovation Japan (SIJ), born from a crowdfunding campaign. It was the first water refill application platform in Japan, and it strives to drive social good through its social missions divided into two aspects: reduce the usage of PET bottles through the water refill of reusable bottles and drive behavioral change in society through a social movement for responsible consumption. The mymizu smartphone app is an open-source map with public water refill spots and mymizu refill spots (businesses such as shops, cafes, restaurants, etc.) that allow people to refill their water bottle for free instead of purchasing PET bottled drinks for single use and then throwing them away. This is extremely pertinent to a society that is becoming increasingly aware of environmental issues but still is currently lacking in engagement. Japan is the 2nd-largest generator of plastic packaging waste per capita, with 30 billion plastic bags used every year and 23 billion PET bottles bought every year, as of 2020 [59]. Hence, its social mission to drive towards a circular economy is crucial, as the environmental reality is that recycling is not the solution to such issues. This makes a holistic approach necessary for the elimination of waste and pollution, the circulation of materials and products, and the regeneration of nature, in order to decouple the consumption of limited resources from economic activity. In late 2022, mymizu also launched an open-source web platform (currently in beta version), created by the community for the community.

From the viewpoint of the BM, mymizu is built around its social mission as its value proposition, which also defines its SI: to raise awareness about environmental issues with PET bottles as a reference and drive social change through behavior for responsible consumption. Mymizu has two sets of "customers" (app users and organizations) and various revenue streams. The users generate revenue only by purchasing mymizu branded items or taking part in the monthly supporter campaign. Refill partners do not generate revenue, as they represent more of a win–win situation by raising awareness of each other and driving foot traffic. The organizations (companies, universities, city governments, etc.) make up much of the current revenue model through partnerships and various services: workshops, seminars, educational activities, talks, consulting for joint product development or communications, brand collaborations, and "mymizu challenges". The mymizu challenge is a paid service consisting of a friendly internal competition, interactive workshops, and lectures, with the aim to raise awareness and to result in behavioral change towards responsible consumption amongst organizational members. As a social startup striving to be financially sustainable while pushing for its social mission, it also relies on NGO-like revenue streams: awards from social business competitions and programs and grants and donations from individuals and organizations (corporate, governmental, etc.).

The marketing is built on creativity, innovation, digital technologies, and society in order to enable a movement with strong storytelling instead of a product or service. The network of refill partners (shops, cafes, restaurants, etc.) are major ambassadors of the mymizu brand, as they raise awareness of mymizu through visuals and conversations. Mymizu's partnerships and collaboration with renowned companies and cities such as Audi, Mitsubishi Chemical Cleansui Corporation, Meisui, LIXIL, Johnson & Johnson, Kobe City, etc., boost the growth of their brand image through awareness of mymizu and its social movement. Furthermore, mymizu also works closely with "mymizu ambassadors": athletes and actors with strong connections to their communities who are inspiring people and changes through their stories and actions. All of these efforts become part of mymizu's narratives for responsible consumption.

### 3.3. Data Collection

For this case study, both primary and secondary data were collected from four sources, ranging from 2019 to 2023: direct interviews with key personnel of mymizu, fieldwork observations, online publications, and social media. In order to ensure the reliability and validity of these data, this research utilized method triangulation for the comparison and

combination of information, which provided a complete picture of mymizu as the case study [60–63]. The collected data are mostly qualitative in nature for understanding the phenomena of the SSI through an in-depth exploration and analysis of people's perspectives and narratives (mymizu's key personnel and users) on the SSI factors. Quantitative data from online publications were also used to analyze mymizu's performance and SI through various indicators.

The primary data were collected via semi-structured interviews, made up of two sessions of about forty-five minutes each, with Mr. Robin Lewis, the co-founder of mymizu. The first preliminary interview was conducted virtually in December 2020 to lay the groundwork by understanding mymizu's social mission. The follow-up interview was conducted virtually in March 2022 to explore mymizu's latest developments, the transitions in its BM, and the status of its SI. This interview helped define the areas to be explored in relation to the BM and the SSI through a deeper understanding of mymizu and its operations [54,64]. Given his leadership role in the organization, the interviews were used to explore his own perspectives for detailed insights into SE, mymizu's current BM, and its forthcoming transitions. The fieldwork observations were carried out at three mymizu-related events as organized by mymizu and SIJ (one talk at a university and two social entrepreneurship pitching and mentoring events) during the months of January, October, and December in the year of 2021. Each observation session lasted from thirty minutes to one hour. These fieldwork observations were useful in gathering descriptive analysis data and gaining new insights from Mr. Robin Lewis and the key personnel of the mymizu team with regards to its philosophy and development, while also serving triangulation purposes [60]. Document studies were used extensively as a reliable source for secondary data, with both qualitative and quantitative data collected from online publications from mymizu, newspapers, and social media throughout the data collection period, with contents from press releases, reports, advertisements, event programs, and company websites. The qualitative data provided a means to track change and development throughout the four years, while the quantitative data from the document studies were used to assess mymizu's SI and components of its BM. Document studies are a highly applicable method for case studies, as they serve as sources of empirical data, providing rich descriptions of unique phenomena, the organization, or an event, all within the context of study, as well as being a popular means of triangulation [65].

### 3.4. Data Analysis

The collected data were investigated through a qualitative content analysis, as this methodology provides knowledge and understanding of the phenomena under study, which is essential for the exploratory nature of this paper [66,67]. For this study, it was an appropriate approach and a useful tool to discover and describe the organizational and social focus, to identify common themes, and to make inferences that can be corroborated with other data and the literature review [66]. In particular, we used the latent projective content analysis method to dive into the implied meanings, developing a deeper comprehension of the factors that enable the SSI through a systemic process of interpretation, all in consideration of SE's context and existing theory on the BM and the SSI [68]. Furthermore, given the rich descriptions of the phenomena, particularly from the document study, a qualitative content analysis enables the data reduction necessary to focus on relevant aspects [69].

Figure 1 summarizes the latent projective content analysis conducted on qualitative data collected from the interviews, fieldwork observations, online publications, and social media. This table brings out the key themes repeated throughout this paper and also provides the underlying reasons behind the numbers seen in Figure 2.

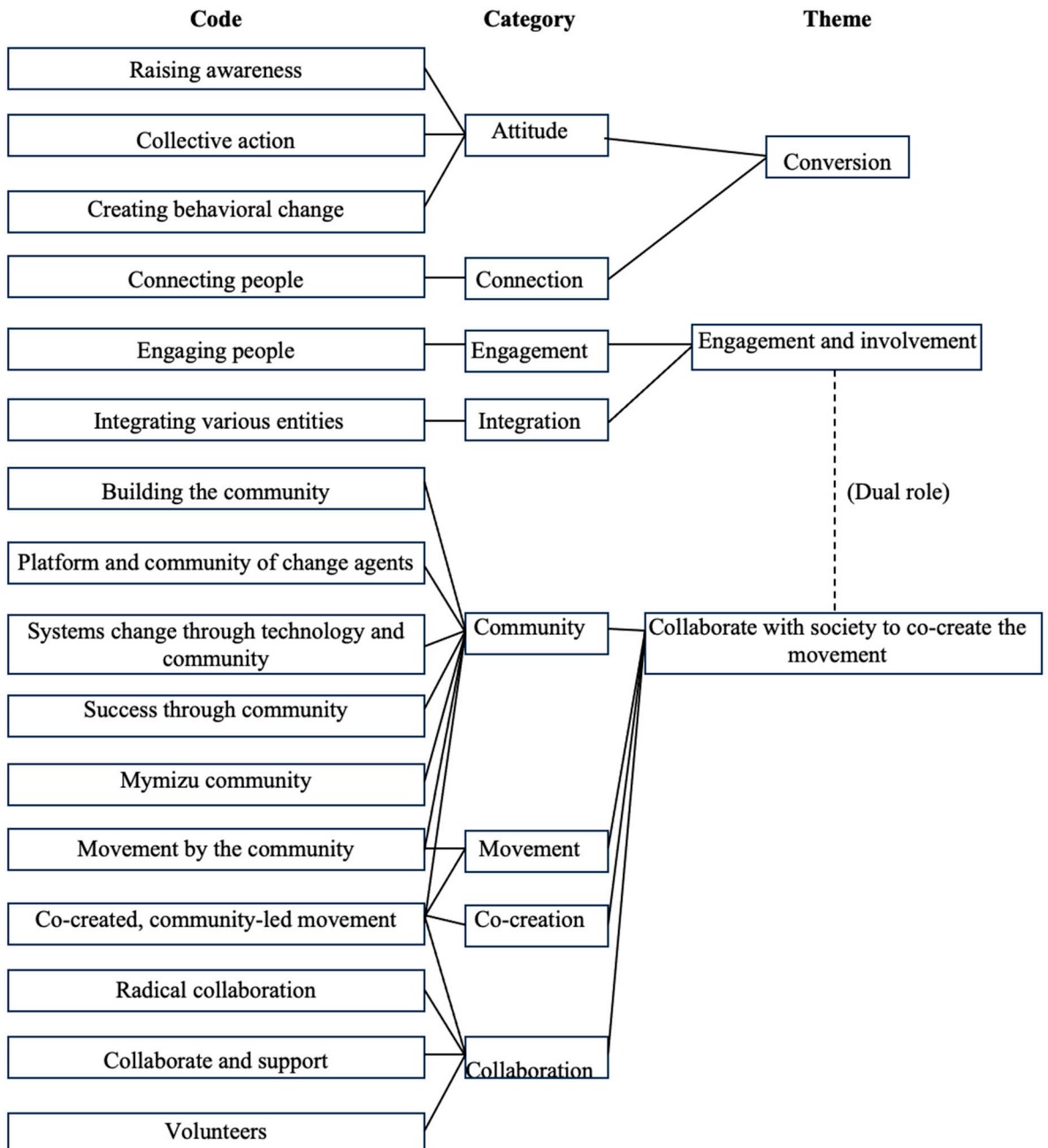

**Figure 1.** Latent projective content analysis.

The figure below (Figure 2) shows the outcomes of mymizu through quantitative data collected since its establishment in 2019, measured at three different points of time (2019–2021–2023) and in between each period, which highlights how mymizu's SI has been scaled through time.

| | 2019 | 2021 | 2023 |
|---|---|---|---|
| **Number of team members** | 8 core members | 8 core members (30+ including volunteers) | 8 core members (30+ including volunteers) |
| **Number of users** | 10,000+ worldwide | 160,000+ worldwide | 200,000+ worldwide |
| **Number of mymizu refill spots** | 160+ in Japan | 1800 worldwide | 2300+ in Japan and 10+ countries |
| **Number of public refill spots** | 8000+ in Japan | 9500+ in Japan 200,000+ worldwide | 10,000+ in Japan 205,000+ worldwide |
| **Number of PET bottles saved** | 100,000+ worldwide (in early 2020 when tracker was launched) | 250,000+ worldwide | 650,000+ worldwide |
| **Number of media coverage** | 30+ | N/A | 500+ worldwide in 20+ countries |
| **Number of participants in events, workshops and programs** | N/A | 40,000+ | 50,000+ |

**Figure 2.** Outcomes of mymizu in 2019, 2021, and 2023.

## 4. Results

Based on the data collected and the analysis, the results showed the impact on the acquisition and conversion of new users through collaboration amongst different stakeholders on the initial phase of the BM, as well as the exponential scaling effect of the SI due to the new dual role taken on by society.

### 4.1. Converting Awareness into Users with Behavioral Change

The first finding is that mymizu has heavily collaborated with various stakeholders, creating a network of partners and collaborators that spreads the mymizu brand across sectors and demographics, going beyond the network of mymizu refill partners, which are considered as their brand ambassadors. In turn, this resulted in raising the awareness of environmental issues and the circular economy, particularly in relation to PET bottles, and inculcating a behavioral change towards the practice and lifestyle of reduce, reuse, and recycle of plastic bottles to design out this waste. Through this journey, stakeholders became able to take action as mymizu users with responsible consumption. As such, it can be inferred that mymizu's goal of driving behavioral change at scale can be achieved by acquiring new users through collaborations at scale.

Since mymizu's launch with its marketing built on creativity, innovation, digital technologies, and society, it established partnerships and collaborations with numerous entities on various scopes: Audi, Cleansui, Meisei High School, IKEA, Kameoka City, Kobe City, LUSH, PADI, etc. This served to spread awareness internally amongst individuals within their organizations, enabling them to work together on joint product development, communications, and brand collaborations. Furthermore, mymizu could also leverage these organizations' brands and communications and spread awareness through them, in addition to the awareness generated through the mymizu refill partners. These partnerships and collaborations also extended to the individual level by working with mymizu ambassadors, athletes, and actors, whose networks further spread awareness.

This impact is put forward by the increase in the number of users between 2019 and 2023, scaling from more than 10,000 worldwide to more than 200,000 around the world,

while the number of mymizu refill partners went from more than 160 in Japan to more than 2300 in Japan (in all 47 prefectures, whereas it began in Tokyo) and worldwide. These numbers show that awareness led to behavioral change and collective action, as well as a further increase in the actions taken in order to spread awareness by being part of the movement. Users had a unanimous viewpoint: "I have been able to reduce so much plastic bottles [...] It is a really great app that is helping our planet" [70].

The number of PET bottles "saved" indicates the number of bottles gone unused thanks to the equivalent amount of water being used to refill from public or refill spots. Between 2020 (when the tracker function was enabled) and 2023, there was an increase from more than 100,000 to more than 650,000 saved worldwide. This highlights the change in behavior, specifically by reducing the consumption of bottled water and by (re)using reusable water bottles: "This is so useful because I don't have to buy bottled water anymore! I used to buy from the vending machine even if I had a reusable water bottle with me because I would eventually run out of water. I didn't know where to refill my bottle but this app solves that issue" [70].

*4.2. Dual Role for Individuals: User and Collaborator*

The second finding is that said users can go beyond their roles and become individual collaborators themselves, driven by a social purpose, and become engaged and involved in mymizu's social mission. In fact, this began at the very early stages of mymizu through crowdfunding that enabled the establishment of mymizu, and throughout its existence, mymizu has worked in close collaboration with many of its users through three collaborative roles: (i) management volunteers, (ii) monthly financial supporters, and (iii) tech collaborators. Such activities tie back to mymizu's mission and philosophy: "co-create an unstoppable movement for sustainability, one bottle at a time" [71] and "The future is co-created. If we can connect millions of mission-driven people, we can kick start a movement and build a world where sustainable living is the norm. That's why we're building a platform and community of change agents" [71]. As such, this resulted in collaborators participating in mymizu's mission.

(i)    Management volunteers

Management volunteers (on a pro bono basis) were essentially part of the mymizu team itself, involved in the human resources of the social startup, holding roles that encompass marketing, communication, UX design, mymizu refill spots and partner networks, product management, etc. Working alongside mymizu's core team, they supplemented much of the needed human resources for mymizu's scaling and the SSI. As mentioned on the mymizu website: "due to high demand" to become a volunteer [72], we can see how society's members were ready to become part of the collaborating workforce.

(ii)    Financial supporters

Financial supporters were new roles that became available through the Monthly Supporter Campaign launched in late 2022. In addition to donations, the monetary supporting actions "contribute to covering towards essential costs" [73] for mymizu to continue their commitment of "keeping mymizu free-of-charge (for both users and refill partners)" [73]. On a monthly basis, this could create a stable source of revenue generation.

(iii)    Tech collaborators

While mymizu had management volunteers in specific tech roles as well, the tech collaborators became part of a tech community when mymizu went open source for its new mymizu Web App (currently in open-source beta). With "Technology + Community = Systems Change" as one of its core beliefs, mymizu launched this project through a hackathon in collaboration with Code Chrysalis (Japan's only Silicon Valley-born coding bootcamp), bringing in the tech community from Tokyo and beyond. This enabled "radical collaboration through technology" in order to "change attitude towards sustainability at scale" [74,75].

## 5. Discussion

In fact, both findings can be interpreted as a single outcome in a loop: partnerships and collaborations with organizations created more awareness, brought new users, and created behavioral change, with users becoming engaged and involved as individual collaborators. This created a virtuous cycle, highlighting the importance of the system through partners and collaborators (both organizations and individuals). Based on these findings, we derived a major discussion point that contributes to the existing literature of SE, the BM, and the SSI, building on an existing theoretical framework.

Han and Shah [15] showed that an organization should rely on and leverage both organizational-level and systemic-level factors, and their inter-relationships, to the SSI instead of focusing only on organizational growth as a means to achieve the SSI. These factors included the process of scaling (the organization, strategies, technology, and data), financing, government policy, and institutional infrastructure. Although the SSI's definition included society as the place where SI takes place [47], the current literature did not explore society (encompassing communities and individuals) as a potential partner and collaborator in this ecosystem.

However, this study shows that society is integrative in a participatory manner [16], with the actors to be termed as "participatory stakeholders" (PS) under various forms on a prolonged basis. The understanding of PS can be supported by both the stakeholder theory and boundary spanner theory. The stakeholder theory explains that stakeholders have a great likelihood to provide important resources that can enable greater efficiency and innovation [76], and that a firm's network of stakeholders can be a source of sustainable competitive advantage [77]. The boundary spanner theory defines personnel who are boundary spanning as "key representatives who engage in various activities on the boundary of an organization" by facilitating both the exchange of information and organizational responses with the external environment. This leads to effective cooperation and problem solving, benefiting the organization and building favorable relationships [78].

As seen in the case study of mymizu, the ecosystem transformed the users to become the PS through involvement and engagement within the BM that positively impacted the organization, while giving a purpose and new roles to these individuals: management volunteers, financial supporters, and technological collaborators. Management volunteers supported mymizu's human resources without specific engagement in terms of contract [10]. Their roles enabled digital entrepreneurial narratives through collaborative communication that generated socially recognizable values shared by both parties, influencing awareness and the target audience's acceptance of social entrepreneurs and ultimately enabling the potential for user interaction and involvement [12]. Financial resources are one of the most pressing challenges in SE, and mymizu's case study also showed the need to diversify and create a stable source of revenue stream. Therefore, the financial supporters represented an unconventional funding stream from within the ecosystem that supports the social startup [5,10]. This also resulted from cognition built through the digital entrepreneurial narratives, putting forward the importance of social meaning construction [12]. The technological collaborators provided the required resources for successfully launching the mymizu Web App, highlighting that social entrepreneurs need to operate within a given community structure for the network's support [4]. This collaboration also reflected the resource acquisition and value co-creation processes posited by Drencheva et al. [79].

This led us to propose a revised version of the ecosystem of the SSI framework created by Han and Shah [15] in Figure 3, adding society as one of the factors that interacts with the others within this ecosystem, particularly with the organization through society's newly found roles in the BM, thus positively influencing the SSI.

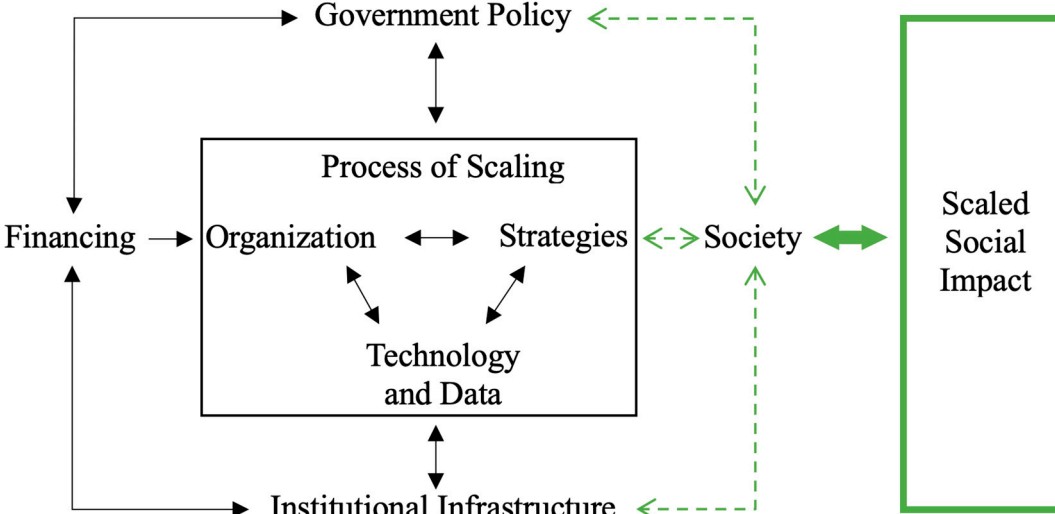

**Figure 3.** Revised ecosystem of SSI.

From a systematic perspective, members of society integrated the revenue streams and the human resources of the organization's BM, transforming external factors into internal resources without being part of the organization and marking an important transition in the BM. This process can then be defined as a regenerative model based on the transitions that influence the SSI, created between the BM and the society, drawing parallel to Demil et al.'s [14] idea of the BM and business ecosystems co-evolving through their interactions. This paved the way for the concept of Regenerative BM (RBM), an increasingly popular research strand [80], which can be understood as having a net positive impact by making the organization's handprint (positive impact created by its product or service) larger than its footprint [81,82]. This also tied back to "how to get $100\times$ the results with $2\times$ the organizations" [44], which was shown by mymizu's results. Finally, with this social startup's impact of reducing the consumption of plastic bottles and changing behavior at large, there was "planetary health and societal wellbeing to nature and society at large", which became its value proposition [83–87]. Thus, mymizu can be inferred to have an RBM, or at least partly, pending further evidence.

**6. Conclusions**

This paper aimed to answer the following research question: what is the role of the BM in the SSI and how does it result in an exponential scaling? The literature review [9–12,15–17] along with mymizu's case study showed that it is important to adopt a holistic approach and involve the ecosystem and its actors. In particular, stakeholders represent an important role and opportunity in the development of SE and the BM, to the SSI. Thus, we conclude that a BM's role in the SSI is to involve key actors and stakeholders of the society within the BM's ecosystem, effectively turning external resources into internal resources by using the value proposition as the shared social meaning construct [12]. Key actors of the society who are engaged are represented as PS, and they may undertake the role and function of management volunteers, financial supporters, and technological collaborators, which lead towards transforming the external factors into internal resources and trigger an RBM.

This study is important for both practitioners and researchers because SE fills in the institutional voids that other entities are unable to respond to. However, in SE, a number of constraints limit the capability to deliver SI, and the SSI is further limited. Hence, when social entrepreneurs innovate the BM by involving key actors of the society, their responses to the socio-environmental issues are scaled, thus working towards greater achievement of the SDGs. The theoretical contribution of this study is the revised framework of the ecosystem of the SSI with the inclusion of society as a component to bring about exponential effects on SI. In terms of the practical contributions for social entrepreneurs, the study

suggests the need to focus on the community surrounding the social mission, to leverage users with shared values from this community, and to include them with roles, functions, and incentives as part of internal resources for resource-scarce organizations. Next, SE should also utilize its communication to increase society's motivation to take on opportunities to become involved. Finally, as a result of this involvement of society, there are greater opportunities to receive support from and collaborate closely with governments or corporations that would support an inclusive society, in order to meet SDGs 11 (Sustainable Cities and Communities) and/or SDG 12 (Responsible Consumption and Production).

However, this paper has some limitations, and further research will be necessary to draw generalizations applicable in other contexts. First, the ecosystem explored in this study works because there is already a thoroughly established and robust ecosystem in Japan: a highly developed economy with strong institutional infrastructure, government policies, and financing. Second, this study is built on the assumption that society is aware of environmental issues but is currently lacking in engagement to tackle such challenges, which is true in the context of Japan in that it is currently lagging behind other similarly developed countries in terms of environmental actions. This is the reason why mymizu was chosen as a single case study for this research, as it provides an ideal context to explore in depth the relationships between the BM and the SSI. Therefore, future researchers should test the revised framework for the ecosystem of the SSI for empirical evidence in other contexts (i.e., social enterprises in different industries, with different types of BM, in different countries, developed and developing, etc.). Cross-national research might also be suggested to draw out differences between different contexts, highlighting the various factors and enablers of the SSI. Finally, given the growing importance of the RBM as a new research strand, further investigation is also encouraged to tie in the SSI to the RBM and find more inter-relationships that can enable a stronger SSI.

**Author Contributions:** Conceptualization, K.K.F. and H.C.G.; methodology, K.K.F.; validation, H.C.G.; formal analysis, K.K.F. and H.C.G.; investigation, K.K.F.; resources, K.K.F. and H.C.G.; data curation, K.K.F.; writing—original draft, K.K.F.; draft preparation, K.K.F. and H.C.G.; writing—review and editing, K.K.F. and H.C.G.; project administration, K.K.F. and H.C.G. All authors have read and agreed to the published version of the manuscript.

**Funding:** The research received 150,000 Japanese yen for Article Processing Charges from Hoe Chin, Goi through the NUCB Business School.

**Institutional Review Board Statement:** Not applicable.

**Informed Consent Statement:** Informed consent was obtained from all subjects involved in the study.

**Data Availability Statement:** Not applicable.

**Acknowledgments:** Special acknowledgement is given to Robin Lewis (co-founder of mymizu), mymizu (Social Innovation Japan, General Incorporated Association), and the NUCB Business School for the cooperation throughout the research implementation and data collection phase and the NUCB Business School and Nagoya University of Commerce and Business for administrative and technical support.

**Conflicts of Interest:** The authors declare no conflict of interest.

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
