# Peer review of "Business Model for Scaling Social Impact towards Sustainability by Social Entrepreneurs"

_sustainability, doi:10.3390/su151814027_

Round 1
Reviewer 1 Report
Dear authors,
Congratulations on this nice piece of work. I applaud your pursuit of this topic.
However, your methodology lacks substance. It is very descriptive. It gives no key indication of how the survey was conducted and you do not justify your method enough.
My question is whether you have ethical clearance from the company for this study?
Going forward, I encourage you to clearly and cogently explain: (1) how the focal study addresses an important industry priority; (2) how the topic is connected to existing theory; (3) what we already know about practice and theory; (4) what specifically we do not know; (5) why we need to know what we do not know; and (6) how this study or inquiry will help close the practical and theoretical gaps between what we know and do not know.
You are expected to address the comments and submit a revised version.
Reviewer 2 Report
This is an interesting paper and I enjoyed reading it. However, there are essential weaknesses that need to be addressed.
1) The introductory/opening section should communicate a little clearer the literature gaps, as well as the study's aims & objectives in order to facilitate the flow of the study.
2) Overall there are good arguments and well researched points made in this paper, but I feel that author needs to take to a further level.
I strongly recommend that you include the following references focused on the target journal and on the paper’s topics:
Goduscheit, R. C., Khanin, D., Mahto, R. V., & McDowell, W. C. (2021). Structural holes and social entrepreneurs as altruistic brokers. Journal of Innovation & Knowledge, 6(2), 103-111. https://10.1016/j.jik.2020.12.001
Méndez-Picazo, M., Galindo-Martín, M., & Castaño-Martínez, M. (2021). Effects of sociocultural and economic factors on social entrepreneurship and sustainable development. Journal of Innovation & Knowledge, 6(2), 69-77. https://10.1016/j.jik.2020.06.001
Mugoni, E., Nyagadza, B., & Hove, P. K. (2023). Green reverse logistics technology impact on agricultural entrepreneurial marketing firms’ operational efficiency and sustainable competitive advantage. Sustainable Technology and Entrepreneurship, 2(2), 100034. https://doi.org/10.1016/j.stae.2022.100034
Zhao, C., Liu, Z., & Zhang, C. (2023). Real or fictional? Digital entrepreneurial narratives and the acquisition of attentional resources in social entrepreneurship. Journal of Innovation & Knowledge, 8(3), 100387. 10.1016/j.jik.2023.100387
3) The research is well-developed.
4) At the end, the author should include clear statements as to where research should now go.
5) Carefully check the references, so as to make sure they are all complete and follow the Guidelines to Authors.
6) Finally, when you submit the corrected version, please do check thoroughly, in order to avoid grammar, syntax or structure/presentation flaws.
Thank you for the opportunity to read the paper.
To be copyedited in the last round of review
Reviewer 3 Report
Of the literary references, hardly half are younger than ten years old, and there are also references from the last century, which I don't know if it is necessary. Personally, I am not a fan of qualitative studies, since they are unrepeatable, and with the complete absence of any statistics, as is the case here, and unverifiable. The authors themselves draw attention to the limitations of the study, giving it a very limited significance - from a cognitive point of view, it was very interesting to read this paper, not so much from a scientific point of view. Thank you.
Reviewer 4 Report
Research GAP: Previously, studies have explained the importance of BM in relation to SE. However, there is a lack of empirical studies on how BM’s transitions through participation of various actors result in SSI.
Japanese social startup, “mymizu”, the first water refill application platform in Japan
The findings show that collaboration amongst different stakeholders on the initial phase of the BM could increase awareness of responsible consumption, convert into actual users for sustainability, and change their behavior. Secondly, members of the society could take on dual roles, both as users and collaborators in the BM, which results in an exponential scaling effect of the social impact.
The contributions of the paper explained well.
Research methodology is so rich and authors used different data gathering types.
“Japan is the 2nd largest generator of plastic packaging waste per capita, with 30 billion plastic bags used every year and 25 billion PET bottles bought every year, as of 2022” I advise the authors should cite this sentence in page 9.
Thank you
Round 2
Reviewer 2 Report
Nothing
Nothing